# Noteworthy trends in maladaptive coping strategies and hindrances to help-seeking behaviour among adolescents living in Malaysia's People's Housing Project (PPR) during the COVID-19 pandemic: A qualitative study

Siti Nur Farhana Harun[1], Noorlaile Jasman[1☉], Feisul Mustapha[2], Norrafizah Jaafar[1], Siti Nadiah Busyra Mat Nadzir[3], Zanariah Zaini[1], Manimaran Krishnan[1], Ponnusamy Subramaniam[4]*

1 Institute for Health Behavioural Research, National Institutes of Health, Ministry of Health Malaysia, Malaysia, 2 Disease Control Division, Ministry of Health Malaysia, Malaysia, 3 Counselling and Psychology Unit, Registrar Office, National Institutes of Health, Ministry of Health Malaysia, Malaysia, 4 Clinical Psychology and Behavioural Health Program, Faculty of Health Sciences, National University of Malaysia, Malaysia

☉ Equal contributing author as first author.
* ponnusaami@ukm.edu.my

## Abstract

### Background

COVID-19 has greatly affected the population, especially those in the low socio-economic group, including residents of the People's Housing Project (PPR) in Malaysia. Adolescents residing in PPR communities are among the most vulnerable groups of young people in urban areas, given their pre-existing conditions of vulnerability, face even greater challenges due to the pandemic. Understanding their mental health and coping strategies is vital to grasp how the pandemic impacts their well-being. Hence, this study aims to explore the coping strategies and barriers to help-seeking behaviour among adolescents living in the Malaysia's PPR communities, focusing on the unique mental health challenges exacerbated by the COVID-19 pandemic. Given the socio-economic vulnerabilities and the heightened mental health challenges during the pandemic, this study provides critical insights into how adolescents in PPR communities navigate psychological distress and mental health support.

### Methods

This qualitative study used a phenomenological research design and was conducted from January to December 2022, involving 47 adolescents aged 10 to 17 years old from 37 PPRs in the Klang Valley. Participants were recruited using the purposive sampling method as this study purposely selected adolescents with moderate, moderately severe,

**Data availability statement:** The data can be requested through National Institutes of Health – Data Repository System (NIH-DaRS) at https://nihdars.nih.gov.my/research/view/142/details. Alternatively, requests can be directed to the Sector for Biostatistics & Data Repository, National Institutes of Health Malaysia, via email at nihdars@moh.gov.my, citing the research ID, NMRR-21-1521-60861 for reference.

**Funding:** Funding provided by UNICEF in the form of an unrestricted grant. The funders had no role in study design, data collection and analysis, decision to publish, or preparation of the manuscript.

**Competing interests:** The authors have declared that no competing interests exist.

**Abbreviations:** GAD-7, General Anxiety Disorder-7; IDI, In-depth Interviews; PHQ-9, Patient Health Questionnaire-9; PPR, People's Housing Project

and severe for PHQ-9 and/or moderate and severe for GAD-7 based on the screening. Participants who agreed to participate were recruited (with consent from parents/guardians) and interviews were set at the participants' convenience. Data were collected using a semi-structured interview guide to conduct the in-depth interviews (IDI). After each IDI session, the recorded interviews were transcribed. Data from the voice recorder were stored on a password-protected computer, and participants' names were replaced with specific codes to ensure confidentiality. The researchers coded all transcripts independently. The transcripts were analysed inductively using a thematic approach to identify recurring themes.

## Results

From the 37 PPRs, 194 adolescents were identified as having psychological distress based on the screening. Among them, 47 agreed to participate in the IDIs, which revealed that these adolescents utilized mainly maladaptive coping strategies, such as avoidance (cognitive distancing, externalization, and internalization), self-harm, vaping, and smoking to deal with stressors related to COVID-19. As for hindrances to help-seeking, three themes were identified such as lack of trust, perceived ineffectiveness of support, and personality.

## Conclusion

Psychological distress among adolescents was prevalent during the pandemic, and they faced hindrances in seeking help. Coping strategies have been identified to help adolescents manage their psychological distress during the pandemic. It is concerning that some had resorted to maladaptive coping mechanisms. These findings emphasized the need for targeted mental health interventions and support systems tailored to vulnerable communities. These interventions could inform policies aimed at strengthening mental health services, fostering better coping strategies and promoting help-seeking behaviours among adolescents in socio-economically challenged communities.

## Introduction

The COVID-19 pandemic has profoundly impacted individuals worldwide, and adolescents are no exception. Adolescents residing in the People's Housing Project or *Projek Perumahan Rakyat* (PPR) in the Klang Valley have encountered distinctive challenges as they struggle with the pandemic within their already vulnerable living conditions. Klang Valley is a metropolitan area in Malaysia that includes the capital city, Kuala Lumpur, and its surrounding areas in the state of Selangor. It is the most developed and densely populated region in Malaysia. The PPR is an initiative by the government aimed at providing affordable housing to low-income families. It is part of Malaysia's broader effort to address housing needs for its citizens, particularly those in urban areas where housing costs are high.

The dense, crowded and low-resource settings predominantly experienced in PPR communities, often means that children in these living environments are more susceptible to socio-economic deprivations, and are at higher risk of developing mental health conditions. Additionally, financial constraints, low literacy levels, poor mental health awareness, and limited access to support services prevent them from seeking help. The COVID-19 pandemic has intensified these challenges, disproportionately impacting low-income families in PPR

communities as identified by the UNICEF's 2021 Families on the Edge (FoE) report. The FoE also reported that one in two parents believed their children's mental state had been affected by the Movement Control Order (MCO) [1]. The pandemic has disrupted the daily routines of these adolescents, including their education and social lives, leading to increased stress, anxiety, and depression. Some symptoms, such as feelings of sadness, loss of interest and pleasure in activities, as well as disruptions to regulatory functions like sleep and appetite, may be exacerbated during the pandemic, due to social isolation resulting from school closures and physical distancing requirements [2]. Additionally, they may experience heightened concerns about the health of their family and friends due to COVID-19, potentially intensifying their anxiety [3]. Adolescents are susceptible to experiencing behavioural and mental health issues, and these challenges may become more pronounced amid the COVID-19 pandemic [4]. Adolescents face an increased likelihood of developing heightened levels of depression and anxiety when subjected to enforced isolation [5]. On a global scale, pooled prevalence estimates of clinically elevated depression and anxiety symptoms were 25.2% and 20.5%, respectively [6].

The mental and behavioural well-being of adolescents can be influenced by a combination of stressful events and variations in coping resources and strategies [4]. Assessing the levels of anxiety, depression and coping mechanisms in adolescents during the pandemic is crucial, given that this period is pivotal for their growth and development [7]. When individuals face significant stress, they might struggle to effectively manage contextual demands, leading them to resort to maladaptive coping strategies. These maladaptive strategies, when prevalent, can manifest as psychopathological symptoms such as stress, anxiety, and phobias [8]. Maladaptive coping strategies may also manifest in response to challenging situations; for example, some individuals may exhibit excessive worry or nervousness when dealing with stress [9] while others may attempt to evade or ignore problems [10] or even detach from their emotions and thoughts as a means of coping [11].

General mental health indicators such as the presence of adaptative coping strategies may improve individuals' enduring, resistance, and adaptation during their life including pandemic context [12]. Utilizing these adaptive coping strategies may also correlate with a greater likelihood of seeking help to manage the pandemic's impact on their mental well-being. Past studies have identified that most young people who utilize maladaptive coping strategies such as self-harm do not reach out for professional help [13]. Some do not even get medical help when they hurt themselves. Even if they have a mental health diagnosis, they might not ask for help. This shows the need to encourage those who are struggling with these feelings to seek help.

Adolescents with anxiety disorders often turn to friends and family to feel better, however, some opt not to seek assistance because they believe they can manage their anxiety on their own or do not perceive the need for external help. They might also feel like nobody cares about their issues or that there are not enough resources to help them [14]. Using mental health literacy interventions is a useful way to encourage better attitudes towards seeking help for depression [15]. The COVID-19 pandemic greatly affected mental health and healthcare accessibility, which in turn impacted how people seek help. Even those already vulnerable to mental health challenges encountered great difficulty in reaching out for support. Additionally, in the context of Asian nations and low and middle income countries, cultural factors such as stigma surrounding mental health issues [16], familial support systems [17], and the level of awareness of mental health [18] may further shape adolescents' coping strategies and help-seeking behaviours. Addressing these cultural factors is critical to understanding and effectively intervening in mental health challenges faces by the adolescents.

Therefore, the main objective of this study is to explore coping strategies among adolescents living in the PPRs during the COVID-19 pandemic, and hindrances to help-seeking

behaviour. This study aims to answer the research question on how adolescents in Malaysia's PPR communities navigated challenges through various coping strategies and what hindered them from seeking support. Understanding these dynamics can provide valuable insights into how best to support adolescents during times of crisis. This research focuses on promoting adaptive coping strategies among underprivileged adolescents in urban poor settings in Malaysia. To the best of our knowledge, there are limited studies on coping strategies in this specific population, which highlights the importance and uniqueness of this study. This research addresses a significant gap in the literature by providing detailed insights into the coping mechanisms of adolescents facing extreme socio-economic challenges during the pandemic, thus underscoring the necessity for targeted mental health interventions and support systems in this vulnerable community.

## Materials and methods

This qualitative study utilized a phenomenological research design and was conducted from January to December 2022. The research preparation began in January and extended until March 2022. Data collection commenced on April 1, 2022, and concluded on September 30, 2022. Data analysis and writing were conducted concurrently with data collection, continuing until December 2022. Participants were selected from 37 PPRs in the Klang Valley, and among those who scored moderate, moderately severe, and severe in the Patient Health Questionnaire-9 (PHQ-9) and/or moderate and severe for the Generalised Anxiety Disorder (GAD-7) during the screening. Adhering to the guidelines set by the Malaysian Medical Research and Ethics Committee (MREC), researchers are ethically obligated to ensure that adolescents experiencing psychological distress or showing signs of self-harm or suicidal thoughts during screening are promptly referred to the identified health clinics. Throughout the data collection process, a psychologist or counsellor was available for immediate support.

During each IDI, either a counsellor or a psychologist was present with the main interviewer. Following each interview session, participants underwent a debriefing session conducted by a counsellor or psychologist. Additionally, a standardized referral letter template was utilized for referrals. Before adolescents were referred, researchers provided intervention materials to be used prior to their healthcare facility visit. These materials, developed in collaboration with Malaysia's National Center of Excellence for Mental Health under the Ministry of Health Malaysia, included tips for mental health care, effective stress management strategies, and contact numbers for immediate assistance if required.

A semi-structured interview guide was developed based on the Transactional Theory of Stress and Coping Model [19] and literature reviews as a guide to conduct the IDIs (S1 File). The Transactional Theory of Stress and Coping Model informed the structure of our interview guide, particularly around primary and secondary appraisals of stress, as well as the use of problem-focused and emotion-focused coping strategies. Primary appraisals involved assessing whether situations were perceived as positive, threatening, or challenging. Secondary appraisals considered the availability of social and psychological resources for coping. The interview guide was reviewed by a clinical psychologist from the National University of Malaysia and public health physicians from the National Centre of Excellence for Mental Health, Ministry of Health Malaysia. In this study, reflexivity was used as a strategy to reduce researcher bias by reflecting on how personal perspectives, assumptions and beliefs could influence data analysis and interpretation. Debriefing and research team member checking were employed as tools to critically examine the researcher's role and potential biases throughout the research process.

This research has been registered under the National Medical Research Registration (NMRR), with ethics approval from the Medical Research Ethics Committee (MREC), Ministry of Health Malaysia on 24 November 2021 [Ref:(17) KKM/NIHSEC/ P21-1528].

## Data collection

The researchers utilized the PHQ-9 and GAD-7 screening tools to assess the mental health status of the participants. Upon receiving the completed screening forms, the researchers carefully assessed and identified individuals who scored within the moderate, moderately severe, and severe ranges on the PHQ-9, as well as within the moderate and severe ranges on the GAD-7. These selected participants were then invited to participate in an IDI session. The researchers provided an explanation of the purpose and process of the IDI to both the parent/guardian and the adolescent. If they agreed and gave consent, an appointment was scheduled at a mutually convenient time. The IDI sessions were conducted in a private room located in the community hall of the respective PPR and lasted approximately 45 minutes to an hour. To ensure the well-being and comfort of the participants, a psychologist or counsellor was present during each session. With the participants' consent, each IDI session was audio-recorded. Participants scoring in the moderate, moderately severe, or severe range on the PHQ-9 and/or moderate and severe scores on the GAD-7 were referred for psychological support at the nearest identified healthcare facility.

## Data analysis

Following each IDI session, interview transcripts were transcribed verbatim. The transcription process was carried out promptly to ensure that no information was omitted and to allow new insights to inform subsequent IDI sessions. The data obtained from the voice recorder were securely stored on a password-protected computer solely for research purposes. To safeguard participant privacy, their identities were kept confidential, and their names were replaced with unique codes. All transcripts were imported into NVivo version 12 and data analysis was conducted using NVivo version 12 software to facilitate systematic coding and organization of themes. Each transcript was independently coded by multiple researchers to ensure comprehensive coverage of the data. A thematic approach by Braun and Clarke [20] was employed to analyse the transcripts and identify recurring themes through an inductive process. Any disagreements in coding were resolved through research team consensus, and whenever discrepancies arose, the researchers worked collaboratively through discussions to solidify a coding framework. Within these discussions, the research team also identified and incorporated new keywords or phrases into the framework. Additionally, the research team consolidated all codes into a codebook, which functioned as a reference throughout the coding process. The research team engaged in further discussions to develop and explore emerging themes. An inter-coder reliability check was performed to enhance the consistency and reliability of the analysis process.

## Results

From the 37 PPRs, a total of 1,578 eligible participants' ages ranged from 10 to 17 years (mean = 13.52, SD = ±2.193), with females 50.5% and males 49.5%, completed the questionnaire. Based on the total number of eligible participants, 194 adolescents were identified as having psychological distress based on the screening. Among them, a total of 47 participants agreed to participate in the IDIs. Data saturation was achieved after 42 interviews, where no new information emerged across identified themes. Five additional interviews were conducted as a confirmatory measure reaffirming that saturation had indeed been reached. In many studies,

even after claiming saturation, additional data is collected to verify this claim [21]. Among the participants, there were 33 females and 14 males, indicating a higher representation of female participants. The distribution of participants across various age groups revealed a greater presence of individuals from the older age groups. In terms of ethnicity, the majority of participants (n = 45) identified as Malays, while there were two participants of Indian ethnicity, and no Chinese ethnicity participants. Table 1 summarizes the sociodemographic profile of the participants.

## Maladaptive coping strategies

During the COVID-19 pandemic, many adolescents experienced its negative impact, and this study found that adolescents in the PPR utilized predominantly maladaptive coping strategies when dealing with psychological distress. Specifically, the research uncovered avoidance as a prevalent coping approach among these adolescents, encompassing sub-themes such as cognitive distancing, externalization, and internalization. Additionally, the study identified other maladaptive coping mechanisms employed by the participants, including self-harm and the use of vaping and smoking as means of managing their distressing experiences. These are summarized in Table 2.

**Avoidance.** This study uncovered a prevalent pattern among participants; the utilization of avoidance as a coping strategy. When confronted with challenging situations or distressing emotions, individuals in this study demonstrated a tendency to employ avoidance such as Cognitive Distancing, Externalization, and Internalization as their chosen approaches to cope.

**Avoidance: Cognitive distancing.** Participants in this study demonstrated a cognitive distancing, where they perceived their problems as relatively unimportant, leading them to convince themselves that they could successfully adapt to their current circumstances. This cognitive distancing mechanism allowed participants to detach themselves emotionally from the perceived significance of their problems. Participants adopted a mindset that downplayed the significance of their challenges, enabling them to navigate the situation with a sense of ease.

*Because I don't think that thing is that important, I can still adapt.*

(R8, female, 16 years old)

**Avoidance: Externalization.** In this study, participants exhibited maladaptive coping strategies when faced with distress, particularly through the act of externalization. When overwhelmed by anger, participants described they resorted to destructive behaviours such as, throwing things, punching walls, or using a knife to scratch surfaces, tear pillows, or damage clothing.

*If feel angry, just punch the wall... to release the anger... use a knife to carve on the wall... sometimes tear the pillow or clothes.*

(R25, female, 17 years old)

One participant explained that when she was extremely angry, she would throw objects without being fully aware of her actions. This lack of control led to unintended consequences, such as accidentally hitting her mother with a thrown object. She expressed that these actions were not deliberate attempts to cause harm but rather impulsive reactions to intense emotional states.

**Table 1. Sociodemographic profile of the participants.**

| ID | Age (years) | Gender | Psychological Distress | |
|---|---|---|---|---|
| | | | Depression | Anxiety |
| R1 | 14 | Male | Moderate | Moderate |
| R2 | 17 | Female | Moderate | Moderate |
| R3 | 16 | Female | Moderate | Mild |
| R4 | 14 | Female | Moderately severe | Severe |
| R5 | 17 | Male | Moderate | Normal |
| R6 | 11 | Female | Moderately Severe | Moderate |
| R7 | 10 | Female | Moderate | Moderate |
| R8 | 16 | Female | Moderate | Normal |
| R9 | 17 | Female | Moderate | Moderate |
| R10 | 12 | Female | Moderate | Normal |
| R11 | 15 | Male | Moderate | Normal |
| R12 | 15 | Female | Moderate | Normal |
| R13 | 13 | Female | Moderately severe | Severe |
| R14 | 16 | Male | Moderate | Mild |
| R15 | 13 | Female | Mild | Moderate |
| R16 | 16 | Male | Moderate | Mild |
| R17 | 13 | Female | Mild | Moderate |
| R18 | 17 | Female | Severe | Severe |
| R19 | 16 | Female | Moderate | Moderate |
| R20 | 16 | Male | Moderate | Moderate |
| R21 | 15 | Female | Moderately severe | Moderate |
| R22 | 14 | Female | Moderate | Mild |
| R23 | 17 | Male | Moderate | Mild |
| R24 | 16 | Female | Mild | Moderate |
| R25 | 17 | Female | Mild | Moderate |
| R26 | 11 | Female | Mild | Moderate |
| R27 | 11 | Male | Moderate | Moderate |
| R28 | 14 | Female | Moderate | Moderate |
| R29 | 13 | Female | Moderate | Moderate |
| R30 | 15 | Female | Moderately severe | Moderate |
| R31 | 16 | Male | Moderate | Mild |
| R32 | 13 | Female | Moderate | Mild |
| R33 | 12 | Female | Moderate | Mild |
| R34 | 17 | Female | Severe | Moderate |
| R35 | 14 | Female | Mild | Moderate |
| R36 | 15 | Male | Moderate | Mild |
| R37 | 13 | Female | Moderate | Mild |
| R38 | 13 | Male | Moderate | Mild |
| R39 | 17 | Female | Moderately severe | Severe |
| R40 | 13 | Male | Moderate | Mild |
| R41 | 14 | Female | Moderately severe | Mild |
| R42 | 17 | Male | Moderate | Mild |
| R43 | 13 | Female | Moderate | Mild |
| R44 | 17 | Female | Moderate | Severe |
| R45 | 14 | Female | Moderate | Mild |
| R46 | 16 | Male | Moderate | Mild |
| R47 | 14 | Female | Moderately severe | Mild |

**Table 2. Summary of themes and supporting quotes.**

| Theme | Description | Supporting Quotes |
|---|---|---|
| **Maladaptive coping strategies** | | |
| Avoidance: Cognitive distancing | Adolescents used strategies like mentally downplaying the significance of stressors to cope with psychological distress. | *Because I don't think that thing is that important, I can still adapt.* (R8, female, 16 years old) |
| Avoidance: Externalization | Adolescents externalized their emotions by acting out physically, such as punching walls, carving on surfaces, or throwing objects, often leading to harm to themselves or others. | *If feel angry, just punch the wall... to release the anger... use a knife to carve on the wall... sometimes tear the pillow or clothes.* (R25, female, 17 years old) *When I was so angry, I didn't know what I was doing and… I went and threw something and it hit my mother… whenever I got angry, it hurt my mother.* (R9, female, 17 years old) |
| Avoidance: internalization *Emotional suppression* | Adolescents suppressed their emotions or isolated themselves to avoid expressing distress. | *Just keep it to yourself, don't share it with anyone, not even tell your best friend, just hide it.* (R16, male, 16 years old) |
| Avoidance: Internalization *Isolating oneself* | | *Stay still. Sitting alone in the room.* (R35, female, 14 years old) *Sometimes I lock myself up (in the room), and cry, then, sometimes I feel like getting angry but I don't know whom to get angry with.* (R19, female, 16 years old) |
| Self-harm | Adolescents engaged in self-harm behaviours as a way to alleviate emotional distress temporarily. | *Usually barcoding... wanting to hurt yourself... feeling satisfied.* (R41, female, 14 years old) *Cutting my arms... look at YouTube, TikTok... feels less stressed... There is no pain when you're stressed.* (R47, female, 14 years old) *I once swallowed up to 3, 4 Panadol pills at a time and then went to sleep, then didn't eat... then it was like... felt relieved. Instead of enduring all the pain, keeping it to yourself... Because I think it's better to die.* (R39, female, 17 years old) |
| Vaping and smoking | Male participants often used vaping and smoking as a stress relief mechanism. | *Vaping... When I'm stressed like that I vape.* (R20, male, 16 years old) |
| **Hindrances to Help-seeking behaviour** | | |
| Lack of trust | Adolescents hesitated to seek help due to fear of confidentiality breaches and lack of belief in support systems. | *My mother, if I tell her something, sometimes she doesn't believe me, if I am sick, if there's any wrong, she doesn't believe me.* (R43, female, 13 years old) *I don't really trust the school counsellor, er... it's always, if you talk about a problem at school, tomorrow the whole school will know about the problem.* (R19, female, 16 years old) |
| Perceived ineffectiveness of support | Participants doubted the efficacy of help-seeking due to past negative experiences and scepticism. | *It is a mess, it's like sometimes when you tell a story, no one understands. Everything seems to be your fault.* (R39, female, 17 years old) *Because even if he knows, he is not able to help us.* (R41, female, 14 years old) |
| Personality | Adolescents refrained from seeking help to avoid being perceived as weak, often influenced by their role in the family. | *We are the kind of people, keep our emotions to ourselves... Because I am the eldest child, so it is like a big responsibility. Then, when we are sad, I have to show that I'm strong even when I am sad.* (R45, female, 14 years old) |

*When I was so angry, I didn't know what I was doing and… I went and threw something and it hit my mother… whenever I got angry, it hurt my mother.*

(R9, female, 17 years old)

**Avoidance: Internalization.** *Emotional suppression:* Within the context of this study, participants exhibited a maladaptive coping strategy known as internalization, where they chose to suppress their emotions and keep their stress hidden from others. Instead of reaching out for support or sharing their burdens, participants opted to isolate themselves within the confines of their rooms, shedding tears in solitude.

*Just keep it to yourself, don't share it with anyone, not even tell your best friend, just hide it.*

(R16, male, 16 years old)

*Isolating oneself:* A significant pattern arose among the participants, demonstrating their tendency to withdraw and isolate themselves when faced with psychological distress. This behaviour of self-imposed isolation serves as a coping mechanism, enabling participants to create a temporary sanctuary where they can process their emotions privately.

*Stay still. Sitting alone in the room.*

(R35, female, 14 years old)

*Sometimes I lock myself up (in the room), and cry, then, sometimes I feel like getting angry but I don't know whom to get angry with.*

(R19, female, 16 years old)

*Self-harm:* Participants exhibited a distressing maladaptive coping strategy known as self-harm. Several individuals shared their inclination towards inflicting harm upon themselves as a means of finding temporary relief from their emotional pain. This reference to "barcoding" suggests the act of self-inflicted harm, which is often characterized by carving or scratching the skin with sharp objects. Participants also reported that they were encouraged by their friends and exposure to social media to harm themselves as a way of relieving emotional distress and substituting it with physical pain. Participants believed that engaging in self-harm would provide them with a sense of relief and release, offering a brief relief from their overwhelming emotional distress.

*Usually barcoding... wanting to hurt yourself... feeling satisfied.*

(R41, female, 14 years old)

*Cutting my arms... look at YouTube, TikTok... feels less stressed... There is no pain when you're stressed.*

(R47, female, 14 years old)

*I once swallowed up to 3, 4 panadol pills at a time and then went to sleep, then didn't eat... then it was like... felt relieved. Instead of enduring all the pain, keeping it to yourself... Because I think it's better to die.*

(R39, female, 17 years old)

*Vaping and smoking:* The findings of this study shed light on a notable trend observed among the majority of male participants. It was discovered that when faced with stress, many

of these individuals turned to vaping and smoking as coping mechanisms. They expressed their belief that engaging in these activities helped them release their stress and find temporary relief.

*Vaping... When I'm stressed like that I vape.*

(R20, male, 16 years old)

### Hindrances to help-seeking behaviour

**Lack of trust.**  The participants in the study exhibited a significant absence of trust when it came to seeking assistance from either formal or informal sources. Their concerns primarily revolved around the fear of their personal issues becoming public knowledge. For instance, one participant shared about her relationship with the mother, highlighting the lack of trust present in their interactions. Whenever they confide in their mother regarding their well-being or any problems they are facing, they often face disbelief and scepticism. Even when they are genuinely unwell or experiencing difficulties, their mothers fail to place faith in their words.

*My mother, if I tell her something, sometimes she doesn't believe me, if I am sick, if there's any wrong, she doesn't believe me.*

(R43, female, 13 years old)

Similarly, another participant expressed their scepticism towards the school counsellor, citing a lack of trust in the individual. Their apprehension stems from the perception that sharing a problem with the counsellor inevitably leads to the entire school becoming privy to the issue. This fear of widespread exposure hinders their willingness to confide in the counsellor, as they feel it compromises their privacy and could potentially lead to unfavourable consequences.

*I don't really trust the school counsellor, er... it's always, if you talk about a problem at school, tomorrow the whole school will know about the problem.*

(R19, female, 16 years old)

**Perceived ineffectiveness of support.**  Participants harboured doubts about the efficacy of the support they might receive. They expressed scepticism that even if someone were aware of their problems, that person would not be equipped to provide the necessary assistance. This scepticism stemmed from their past experiences, where individuals who were aware of their issues failed to offer effective help or guidance. The participants firmly believed that sharing their problems with others would result in unfavourable outcomes, such as being reprimanded, misunderstood, blamed, or denied the desired and deserved support. Their fear of not being understood and their belief that others were incapable of providing assistance intensified their hesitation and reluctance to seek the necessary support.

*It is a mess, it's like sometimes when you tell a story, no one understands. Everything seems to be your fault.*

(R39, female, 17 years old)

*Because even if he knows, he is not able to help us.*

(R41, female, 14 years old)

**Personality.** The participants exhibited a strong inclination to tackle their problems independently rather than seeking assistance from others, driven by a desire to avoid being perceived as weak individuals. They specifically referred to their role as the eldest child, highlighting the responsibility they felt to be the pillar of strength within their family.

*We are the kind of people, keep our emotions to ourselves... Because I am the eldest child, so it is like a big responsibility. Then, when we are sad, I have to show that I'm strong even when I am sad.*

(R45, female, 14 years old)

## Discussion

To the best of our knowledge, this is the first qualitative study to explore coping strategies among adolescents living in the PPRs in the Klang Valley, Malaysia during the COVID-19 pandemic and hindrances to help-seeking behaviour. When faced with stressful situations, adolescents have the option to utilize either positive coping mechanisms, which are also referred to as adaptive coping strategies, or negative coping mechanisms, known as maladaptive coping strategies. Adaptive coping mechanisms include seeking social support, acceptance, and positive reappraisal, while maladaptive coping mechanisms are characterized by self-destructive tendencies [22]. This study found that adolescents in this study employ significantly more maladaptive coping strategies in response to the psychological distress caused by the COVID-19 pandemic. This includes using avoidance, self-harm, vaping and smoking to alleviate the stress.

As reported in this study, globally it was well established that COVID-19 pandemic increased anxiety and depression among adolescents [23,24]. Our current findings on maladaptive coping strategies are in line with many previous literatures. A one-year follow-up study with 153 students in Italy after the COVID-19 pandemic showed an increase in maladaptive behaviours such as unprotected sex, self-harm ideation, self-harm behaviours, binge eating and aggressive behaviours [25]. Another study with 2,280 Slovak youth showed extreme maladaptive coping strategies which are deliberate self-harm behaviours, and this was prevalent among female adolescents in our study [26]. Furthermore, another qualitative study showed that a significant number of adolescents have a tendency towards adopting maladaptive coping strategies and refused to seek professional support [27], similar to our study.

To elaborate further, avoidance coping can be described as a passive approach where an individual withdraws from a stressor or as an active strategy where the individual redirects their focus away from or attempts to evade the stressor [28]. Another perspective, exemplified in previous research, incorporates selective disregard for unpleasant aspects of events and increased focus on positive aspects to the extent that the problem becomes less prominent in one's awareness [29]. In this study, adolescents utilized avoidance coping, for instance, cognitive distancing, externalization, internalization such as emotional suppression, and isolating themselves, in order to cope with their psychological distress. This finding is in line with a previous study, that people from lower socioeconomic background tend to use avoidance coping strategies during the COVID-19 pandemic [30]. Participants in this study reported engaging in strategies such as denying, minimizing, and distancing themselves cognitively and behaviourally to directly deal with their problems. During the COVID-19 pandemic, this type of maladaptive coping was substantially connected with psychological distress [31]. This was further supported by a study in a diverse population that clearly indicated disengagement coping was linked with a higher level of depression and anxiety due to COVID-19 stressors

[32]. Extreme life hardship can also contribute to suicidal ideation [33], including the negative influence of the COVID-19 pandemic, as evidenced by our qualitative findings. Education and promotion of proactive coping strategies for adolescents from low-income families may be beneficial in assisting them to cope with future life challenges.

Children and adolescents are the most psychologically vulnerable group due to adverse conditions. This includes the impact of COVID-19 on externalization and internalization problems, which have been thoroughly studied in both quantitative and qualitative research [34]. The most prevalent maladaptive coping strategies adopted by study participants were externalization and internalization. In this study, adolescents display externalizing emotions like hostility and anger, which also manifested in behaviours such as aggression and violent. Additionally, internalizing problems during the COVID-19 pandemic were highly evident in the current qualitative findings and consistent with many other qualitative studies, including one conducted in Turkey [35]. Children in this study experienced feelings of depression, suicidal ideation, anxiety, guilt, and fear, all of which contributed to social withdrawal and isolation. These results concur with past studies [36,37] which indicated that both internalizing and externalizing problems in children were more prevalent during the COVID-19 pandemic compared to before the pandemic. This is mainly due to the loss of play activities, social isolation, and many other stressors posed by the COVID-19 pandemic. Prevention of internalizing and externalizing problems in children and adolescents is important to improve psychological well-being and adaptability in their daily routine.

Another study revealed that adolescents often turn to self-harm to regulate their negative emotions [38]. In this study, adolescents expressed a preference for enduring physical pain through self-harm in order to temporarily alleviate their emotional distress. The infamous method used for self-harm involves cutting the skin with a sharp object, such as a knife, colloquially known as "Barcoding". This trend was not only observed among the participants in this study but also occurred elsewhere. Research conducted in Slovakia among Slovak adolescents found that high prevalence of self-harming behaviour. The study suggests that deliberate self-harm serves as a way for young individuals to cope with emotional distress and difficulties they may be facing [26,39].

During the IDIs, participants were asked about the derivation of the behaviour they engaged in. They revealed that they came across it on social media and decided to try it out due to curiosity about its effectiveness in relieving stress. This discovery highlights the significant role played by social media as an influential source of information for adolescents. The findings also shed light on an important aspect which is the potential inability of adolescents in effectively filtering and evaluating the information they encounter on social media. This lack of discernment may expose them to negative or potentially harmful influences, as they might not possess the necessary skills to critically assess the reliability or appropriateness of the information they come across.

Additionally, some participants mentioned that they adopted the same behaviours after seeing their peers at school doing it. Peer observation caused participants' personal preferences to mirror those of their peers [40]. It is widely recognized that peers can exert a significant influence in the lives of adolescents. Thus, there is a crucial need to instil comprehensive knowledge and awareness regarding mental health among adolescents. Recognizing the power and impact of peer interactions, it is important to help adolescents understand and learn how to take care of their mental well-being. By fostering a culture of informed and healthy mental health practices, we can empower adolescents to make informed decisions, seek support when needed, and promote a positive and supportive social environment.

In this study, we discovered that male participants leaned towards using vaping and smoking as a method of stress relief. Stress reduction is often cited as a reason for smoking [41].

The primary motivation behind youth vaping behaviour is often relaxation and relief from stress and anxiety. Vaping nicotine is believed to offer immediate mental health advantages by reducing negative feelings associated with stress, anxiety, and depression. It is possible that young individuals turn to vaping as a coping mechanism and self-medication strategy to manage and improve their emotional wellbeing [42]. Hence, it is imperative to teach adolescents adaptive coping strategies so that they can avoid resorting to negative coping mechanism as their primary means of alleviating stress.

This study found several factors as hindrances to help-seeking behaviour such as lack of trust, perceived ineffectiveness of support and individual's personality. Trust plays a crucial role in adolescents seeking support or help from formal or informal sources. Adolescents faces a challenging situation with respect to help-seeking for mental health problems. They rely on adults for support, yet at the same time seek independence and do not want their parents to know about their problems [43]. They exhibit reluctance in placing their trust in adults when it comes to seeking help with emotional challenges [44]. The matter of privacy and confidentiality emerged as the most important issues among these adolescent [44,45] and there was a significant sense of doubt that confidentiality implied the counsellor would refrain from sharing consultation details with others. As adults, it is important to display empathy and cultivate an environment where adolescents feel comfortable trusting. This approach will contribute to enhancing their willingness to seek help.

Positive attitudes, such the perceived effectiveness of support, are crucial to increasing help-seeking behaviour during the COVID-19 pandemic [46]. However, the outcome of interview sessions with participants indicated that they perceived seeking help from significant others like their mother and father would result in blame, being scolded, and misunderstanding between them. In addition, children and adolescents also perceive that they will not get the expected support which leads them to stop trying altogether. Previous studies have shown that college students' perceptions of social support from family during the COVID-19 pandemic were a major influence on their mental health and overall quality of life [47]. Those students who perceived a higher level of social support from family and friends reported less depression [48]. In contrast, the present study population has a negative perception of the support system, which hindered them from getting help. If an appropriate intervention is not available, this unfavourable perception will eventually result in further poor help-seeking behaviour and more serious mental health issues. Awareness campaigns and promoting helping-seeking behaviour may improve a positive attitude toward seeking help in difficult periods.

One factor that influences mental health help-seeking behaviour among young people is their personality [49]. Although it is premature to discuss participants personality traits in the present study, some characteristics are evident when it comes to hindrance towards help-seeking behaviour. Adolescent participants demonstrate their reluctance to seek for assistance because they are worried that doing so will lead to others labelling them as having weak personalities. In addition, some adolescents wanted their family and friends to view them as a strong character despite their severe internalizing problems. As can be seen in a previous study, personality is a factor influencing coping strategies, including avoidance-oriented coping strategies [50]. For instance, during the COVID-19 pandemic, persons with extroversion personality traits claimed to have experienced greater difficulties as a result of social isolation [51]. According to another study, students who were more open and conscientious were able to cope with the COVID-19 crisis better than those who were less open and conscientious [52]. It is more beneficial to identify and provide support based on an adolescent's personality traits.

The strength of this study offers valuable insights into the complexity of a social phenomenon, providing context-rich information. We were able to capture details of

participants' experiences, perceptions and feelings, enabling us to understand the underlying context, motivations and drivers. The findings also provided contextual understanding, which enabled us to develop a more holistic perspective of the participants' coping strategies and help-seeking behaviour. This study also enabled us to have close engagements with the participants, allowing their voices to be heard and acknowledged. The findings of this study highlighted the urgent need for public health policy makers to prioritize mental health interventions tailored to adolescents in low-resource settings such as PPR communities. Specifically, integrating mental health literacy programs into the school curricula and community-based outreach programs can foster better coping strategies and encourage help-seeking behaviours. Policies should also aim to enhance the accessibility and affordability of mental health services, ensuring they address the unique socio-economic challenges faced by these adolescents.

However, there are few limitations in the present study. Although this study explored coping strategies and help-seeking behaviours among adolescents using in depth qualitative interview, they lacked in generalizability. Methodologically, qualitative studies itself do not provide objective outcomes which can further limit the generalizability of current findings [53,54]. Qualitative studies offer subjective insights based on individual experiences and because this method do not yield quantifiable results, it can be challenging to apply the findings broadly to other contexts or populations.

Secondly, the presence of the researcher and counsellor during the IDIs may affect participants' responses. Thirdly, self-reporting data can provide rich information, participants' perspectives and opinion on specific issues, but biases may occur due to conditions such as selective recalls, social preferability [55] and false memories. Thus, adolescences in this study may cause biases in recalling and reporting their experiences during COVID-19 pandemic.

## Conclusion

This study revealed the prevalence of maladaptive coping strategies among the adolescent population in the PPR, particularly in the face of the psychological distress brought about by the pandemic. The adoption of strategies such as avoidance, self-harm, vaping, and smoking to alleviate stress reveals the gravity of the challenges they were confronted with. Furthermore, this study unveiled a range of hindrances that inhibit adolescents' willingness to seek help during times of distress. The identified factors, including lack of trust, perceived ineffectiveness of support, and individual personality traits, highlight the complex interplay of psychological, social, and environmental factors that shape adolescents' attitudes toward seeking assistance.

As the world grapples with the ongoing long-term effects of the COVID-19 pandemic, it becomes increasingly evident that tailored interventions are imperative to cater to the specific needs of adolescents living in vulnerable communities such as PPRs. The findings of this study emphasize the urgency of implementing support services and interventions that not only address the immediate psychological distress, but also equip adolescents with adaptive coping skills and strategies that foster resilience in the face of adversity.

Future research on adolescents' mental health during and after the COVID-19 pandemic should consider several directions to overcome limitations identified in current studies. Longitudinal studies are essential to track changes in coping strategies and help-seeking behaviours over time, offering insights into long-term effects. Intervention studies tailored to vulnerable communities, like psycho-education and resilience training, can test effectiveness in promoting adaptive coping strategies. Comparative studies between adolescents in different community types such as urban, rural and affluent settings can highlight unique challenges, aiding targeted intervention design. Community-Based Participatory Research (CBPR)

involving adolescents, parents, educators, and local organizations ensures culturally relevant interventions and builds trust.

Additionally, policy evaluation is crucial to identify gaps and improve mental health support and social services for adolescents. Mental health literacy programs can increase help-seeking willingness and adaptive coping through education and stigma reduction however, the local community must be engaged to understand the local issues and challenges in implementing these programs. Lastly, technological interventions like mobile apps and online counselling should be explored for accessible support, with research on their effectiveness and acceptability in vulnerable communities. Addressing these areas can enhance understanding and support for adolescent mental health during crises like the COVID-19 pandemic.

In summary, this study contributes to the growing body of knowledge surrounding adolescents coping strategies and help-seeking behaviour during crises, like the COVID-19 pandemic. The insights garnered from this research highlight the pressing need for a multi-faceted approach, that combines mental health support, community engagement, and policy reform, to create a living environment in which adolescents feel empowered to adopt adaptive coping strategies and seek help when needed. By addressing these challenges head-on, we can pave the way for the holistic well-being of adolescents in vulnerable communities. By highlighting the coping strategies and barriers to help-seeking behaviours, the findings provide valuable insights into how adolescents in low-resource settings, navigate psychological distress. These findings emphasize the critical need for community-based mental health programs tailored to the PPR context. Such initiatives could focus on fostering adaptive coping strategies, reducing stigma around mental health, and improving access to support services. Policy makers and practitioners can leverage these insights to develop targeted interventions that promote mental health and resilience among vulnerable adolescents, ensuring their well-being in both times of crisis and beyond.

## Supporting information

**S1 File. Interview guide.**
(ZIP)

AcknowledgmentThe authors would like to thank the Director General of Health Malaysia for his permission to publish this paper.

## Author contributions

**Conceptualization:** Siti Nur Farhana Harun, Noorlaile Jasman, Norrafizah Jaafar, Siti Nadiah Busyra Mat Nadzir, Manimaran Krishnan, Ponnusamy Subramaniam.

**Formal analysis:** Siti Nur Farhana Harun.

**Funding acquisition:** Siti Nur Farhana Harun.

**Investigation:** Siti Nur Farhana Harun.

**Methodology:** Siti Nur Farhana Harun.

**Project administration:** Siti Nur Farhana Harun.

**Supervision:** Siti Nur Farhana Harun, Feisul Mustapha, Manimaran Krishnan.

**Validation:** Siti Nur Farhana Harun.

**Writing – original draft:** Siti Nur Farhana Harun, Siti Nadiah Busyra Mat Nadzir, Ponnusamy Subramaniam.

**Writing – review & editing:** Siti Nur Farhana Harun, Noorlaile Jasman, Feisul Mustapha, Norrafizah Jaafar, Zanariah Zaini, Manimaran Krishnan, Ponnusamy Subramaniam.

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
