## [Decision Letter · Decision Letter 0]

13 May 2024

PONE-D-23-42611Noteworthy Trends in Maladaptive Coping Strategies and Hindrances to Help-Seeking Behaviour Among Adolescents Living in Malaysia's People’s Housing Project (PPR) During the COVID-19 Pandemic: A Qualitative StudyPLOS ONE

Dear Dr. Harun,

Thank you for submitting your manuscript to PLOS ONE. After careful consideration, we feel that it has merit but does not fully meet PLOS ONE’s publication criteria as it currently stands. Therefore, we invite you to submit a revised version of the manuscript that addresses the points raised during the review process.

We look forward to receiving your revised manuscript.

Kind regards,

Adobea Yaa Owusu, MA, PhD, MPH

Academic Editor

PLOS ONE

Journal Requirements:

Funding provided by UNICEF in the form of an unrestricted grant.

4. In the online submission form, you indicated that The datasets gathered and used in this study are not accessible to the public due to the inclusion of the confidential interview transcripts which necessitates the protection of participants’ anonymity. If additional data is needed, it can be requested through the corresponding author.

7. One of the noted authors is a group or consortium. In addition to naming the author group, please list the individual authors and affiliations within this group in the acknowledgments section of your manuscript. Please also indicate clearly a lead author for this group along with a contact email address.

8. Please amend either the abstract on the online submission form (via Edit Submission) or the abstract in the manuscript so that they are identical.

9. Your ethics statement should only appear in the Methods section of your manuscript. If your ethics statement is written in any section besides the Methods, please move it to the Methods section and delete it from any other section. Please ensure that your ethics statement is included in your manuscript, as the ethics statement entered into the online submission form will not be published alongside your manuscript. 

Additional Editor Comments:

None

Reviewers' comments:

Reviewer's Responses to Questions

**Comments to the Author**

1. Is the manuscript technically sound, and do the data support the conclusions?

Reviewer #1: Yes

Reviewer #2: Partly

2. Has the statistical analysis been performed appropriately and rigorously? 

Reviewer #1: N/A

Reviewer #2: Yes

3. Have the authors made all data underlying the findings in their manuscript fully available?

Reviewer #1: Yes

Reviewer #2: No

4. Is the manuscript presented in an intelligible fashion and written in standard English?

Reviewer #1: Yes

Reviewer #2: Yes

5. Review Comments to the Author

**Reviewer #1:**  In this paper the authors conducted a qualitative study to examine coping strategies and barriers to help-seeking among adolescents living in people’s housing projects during the COVID-19 pandemic.

The vulnerability of the study population adds to the importance of this paper, and the emphasis on promoting adaptive coping strategies among this subgroup is a well-received take-home message.

A strength of the present study is its rigorous standard for study design and procedures, while the manuscript itself is clear and generally well-written.

Below are my specific comments that are aimed at strengthening the paper:

1. In the Introduction, the authors can provide contextual information on PPR and Klang Valley to help readers who are unfamiliar with the context appreciate why the study population are vulnerable and merit scientific importance, and better appreciate the paper’s contribution.

2. p.8, under the Externalization subheading: Describing participants as resorting to destructive behaviours such as hitting their mother sounded like it was done on purpose, which would be misleading in my opinion as R9’s quoted response seemed to suggest otherwise.

The authors may wish to clarify and/or revise accordingly.

3. p.13, under the Discussion section: The authors may need to expound further on the sentence “This trend was not only observed among the participants in this study but also occurred elsewhere.” The specific word “elsewhere” is vague, and here the discussion may be enriched with specific examples and/or citations to explicitly engage with the literature – either for similarly vulnerable adolescents within the Asian Pacific context or other sociocultural contexts.

Minor comments:

- The second sentence under the Methods section is a repeat of the first.

- There seems to be a typo in the second sentence of Strengths and Limitations: “we were able to capture”?

- If relevant, the authors may consider adding interview guidelines and/or structure as supplementary materials.

**Reviewer #2: ** Critical points and suggestions for improvement of the manuscript.

Introduction

1. The introduction does not clearly establish the novelty of the study. It would benefit from a more detailed explanation of how this study fills a specific gap in the existing literature on adolescents' coping strategies in Malaysia or similar settings.

2. The rationale for focusing on adolescents in Malaysia's People’s Housing Project (PPR) during the COVID-19 pandemic is inadequately explored. There's a need for more specific data or previous studies that highlight unique issues faced by this demographic that warrant this research.

Methods

4. The purposive sampling method is described but lacks a detailed justification for why this method ensures a representative sample of the adolescent population in the PPRs, potentially biasing the results.

5. The article mentions a semi-structured interview guide based on specific theories but fails to provide enough detail about the questions asked. This omission limits the replicability of the study.

6. While the study obtained ethics approval and parental consent, it does not discuss how it addressed potential emotional distress caused to participants discussing their coping strategies and mental health, which is crucial given the sensitive nature of the subject.

Results

7. The results section provides an aggregate view but lacks a detailed demographic breakdown that could reveal important patterns or differences in coping strategies among different age groups or between genders.

8. The claim of data saturation is made after 42 interviews, but there is no explanation of how this was determined or why additional interviews were conducted if saturation was achieved.

Discussion

9. The discussion does not adequately compare the study’s findings with existing research. For instance, how do the maladaptive strategies identified align or differ from those found in other cultural or socioeconomic settings?

10. The discussion makes broad generalizations about the prevalence of psychological distress and coping strategies among adolescents without considering the limitations inherent in qualitative research, such as the lack of generalizability.

11. The discussion does not address the methodological limitations, such as potential biases in self-reporting and the impact of the interviewer’s presence on participants' responses.

Conclusion

12. The conclusion emphasizes the need for interventions without a critical examination of the practical challenges or potential barriers in implementing such programs in the PPR context.

13. The paper does not suggest areas for future research or how subsequent studies could overcome the limitations identified in this study.

14. While the study aims to inform policy, it provides vague recommendations for policymakers without concrete steps or consideration of policy implementation challenges.

6. PLOS authors have the option to publish the peer review history of their article (what does this mean? ). If published, this will include your full peer review and any attached files.

**Do you want your identity to be public for this peer review?** For information about this choice, including consent withdrawal, please see our Privacy Policy .

Reviewer #1: No

Reviewer #2: No

---

## [Author Response · Author response to Decision Letter 1]

22 Jul 2024

As attached in the Response to the reviewer document

---

## [Decision Letter · Decision Letter 1]

14 Nov 2024

PONE-D-23-42611R1Noteworthy trends in maladaptive coping strategies and hindrances to help-seeking behaviour among adolescents living in Malaysia's People’s Housing Project (PPR) during the COVID-19 pandemic: a qualitative studyPLOS ONE

Dear Dr. Subramaniam,

Thank you for submitting your manuscript to PLOS ONE. After careful consideration, we feel that it has merit but does not fully meet PLOS ONE’s publication criteria as it currently stands. Therefore, we invite you to submit a revised version of the manuscript that addresses the points raised during the review process.

Please submit your revised manuscript by Dec 29 2024 11:59PM. If you will need more time than this to complete your revisions, please reply to this message or contact the journal office at plosone@plos.org . Please include the following items when submitting your revised manuscript:

We look forward to receiving your revised manuscript.

Kind regards,

Adobea Yaa Owusu, MA, PhD, MPH

Academic Editor

PLOS ONE

Journal Requirements:

Reviewers' comments:

Reviewer's Responses to Questions

**Comments to the Author**

1. If the authors have adequately addressed your comments raised in a previous round of review and you feel that this manuscript is now acceptable for publication, you may indicate that here to bypass the “Comments to the Author” section, enter your conflict of interest statement in the “Confidential to Editor” section, and submit your "Accept" recommendation.

Reviewer #1: All comments have been addressed

Reviewer #3: All comments have been addressed

2. Is the manuscript technically sound, and do the data support the conclusions?

Reviewer #1: Yes

Reviewer #3: Yes

3. Has the statistical analysis been performed appropriately and rigorously? 

Reviewer #1: Yes

Reviewer #3: Yes

4. Have the authors made all data underlying the findings in their manuscript fully available?

Reviewer #1: Yes

Reviewer #3: Yes

5. Is the manuscript presented in an intelligible fashion and written in standard English?

Reviewer #1: Yes

Reviewer #3: Yes

6. Review Comments to the Author

Reviewer #1: I thank the authors for addressing all my earlier comments in the revised manuscript. My concerns have been thoroughly addressed and I am of the opinion that the strengthened manuscript is ready for publication.

Reviewer #3: Reviewer report on noteworthy trends in maladaptive coping strategies and hindrances to help-seeking behaviour among adolescents living in Malaysia's People’s Housing Project (PPR) during the COVID-19 pandemic: qualitative study

1. Abstract

1. Clarity and Structure: The abstract could benefit from clearer wording, particularly in describing the research question and the study's significance. Opening with a concise statement on the specific challenges faced by adolescents in Malaysia’s People’s Housing Project (PPR) communities due to the COVID-19 pandemic would improve focus.

2. Relevance of the Research Question: Adding a sentence on why this study is particularly relevant to PPR communities in Malaysia would strengthen the justification and make the abstract more compelling.

3. Methods: Consider adding a phrase on the sample size (e.g., number of participants) and the specific methods used (e.g., semi-structured interviews), as this provides readers with a clearer understanding of the study's scope.

4. Results :To improve clarity, specify some of the maladaptive coping strategies and barriers identified (e.g., avoidance, lack of trust in mental health services). Including one or two specific examples would make the findings more tangible for the reader.

5. Conclusions and Implications: Strengthen the conclusion by mentioning potential implications for policy or practice. For example, a brief statement about how this research could inform mental health interventions tailored to disadvantaged communities would make the abstract more impactful.

2. Introduction

• To strengthen the relevance, consider providing a brief overview of the unique characteristics of the PPR communities (e.g., socio-economic constraints, limited access to mental health resources). This would contextualize the challenges faced by adolescents in these settings and reinforce the importance of the study.

• While the authors have referenced international studies on adolescent coping and help-seeking, the literature review could benefit from further discussion of any regional or cultural factors specific to Southeast Asia or Malaysia, which might influence coping strategies and help-seeking behaviors. This would provide additional context for the study’s relevance

• Consider explicitly stating the research question or hypothesis at the end of the introduction. This would give readers a clear understanding of the study’s focus and objectives.

3. Methods

• A more detailed explanation of the coding and analysis process would improve transparency. Specifically, it would be beneficial to describe how coding was conducted (e.g., use of software, independent coding by multiple researchers) and how disagreements were resolved to ensure inter-coder reliability.

• Additionally, mentioning whether the researchers used any form of reflexivity to

minimize bias would strengthen the methodological robustness.

4. Results

• Including some quantitative data, such as the proportion of participants endorsing each coping strategy or barrier, would provide additional context and help quantify the prevalence of specific behaviors.

• Furthermore, a summary table outlining each theme with supporting quotes could enhance the visual presentation of the findings, making the results more accessible to readers.

5. Discussion

• The authors could further expand on how their findings might influence specific public health or educational policies, especially in low-resource settings, and suggest avenues for future research

6. Conclusion

• A brief statement highlighting the potential for community-based mental health programs tailored to the PPR context would provide a forward-looking end to the paper, emphasizing actionable insights that policymakers and practitioners can take from this research.

7. PLOS authors have the option to publish the peer review history of their article (what does this mean? ). If published, this will include your full peer review and any attached files.

**Do you want your identity to be public for this peer review?** For information about this choice, including consent withdrawal, please see our Privacy Policy .

Reviewer #1: No

Reviewer #3: **Yes: ** Martina Mchenga

---

## [Author Response · Author response to Decision Letter 2]

24 Dec 2024

Abstract

1. Clarity and Structure: The abstract could benefit from clearer wording, particularly in describing the research question and the study's significance. Opening with a concise statement on the specific challenges faced by adolescents in Malaysia’s People’s Housing Project (PPR) communities due to the COVID-19 pandemic would improve focus.

Thank you for your valuable feedback. We appreciate your observation regarding the relevance of this study to the PPR community. In the abstract we added below sentences to highlight the relevance of the research question as per suggestion.

Line 29-31: “Adolescents residing in PPR communities are among the most vulnerable groups of young people in urban areas, given their pre-existing conditions of vulnerability, face even greater challenges due to the pandemic”

Line 33-36: “This study is to explore the coping strategies and barriers to help-seeking behaviour among adolescents living in Malaysia's People's Housing Project (PPR) communities, focusing on the unique mental health challenges exacerbated by the COVID-19 pandemic.”

2. Relevance of the Research Question: Adding a sentence on why this study is particularly relevant to PPR communities in Malaysia would strengthen the justification and make the abstract more compelling.

Thank you for pointing out this important point. We have highlighted this sentence at the revised abstract in the latest manuscript;

Line 36-38: "Given the socio-economic vulnerabilities and the heightened mental health challenges during the pandemic, this study provides critical insights into how adolescents in PPR communities navigate psychological distress and access mental health support."

This addition strengthens the justification by connecting the study's importance to the specific context of PPR communities and their unique struggles during the pandemic.

3. Methods: Consider adding a phrase on the sample size (e.g., number of participants) and the specific methods used (e.g., semi-structured interviews), as this provides readers with a clearer understanding of the study's scope.

Thank you for your valuable feedback.

We have highlighted this in the revised manuscript;

Line 41-43: “This qualitative study used a phenomenological research design and was conducted from January to December 2022 involved 47 adolescents aged 10 to 17 years old from 37 People’s Housing Project in the Klang Valley.”

Line 47-48: “Data were collected using a semi-structured interview guide to conduct the in-depth interviews (IDI).”

4. Results :To improve clarity, specify some of the maladaptive coping strategies and barriers identified (e.g., avoidance, lack of trust in mental health services). Including one or two specific examples would make the findings more tangible for the reader.

Thank you for your valuable feedback.

We have highlighted this suggestion in the revised manuscript;

Line 57-61: Among them, 47 agreed to participate in in-depth interviews (IDIs), which revealed adolescents utilized maladaptive coping strategies, such as avoidance (cognitive distancing, externalization, and internalization), self-harm, vaping, and smoking to deal with stressors related to COVID-19. As for hindrances to help-seeking, three themes were identified such as lack of trust, perceived ineffectiveness of support, and personality.

5. Conclusions and Implications: Strengthen the conclusion by mentioning potential implications for policy or practice. For example, a brief statement about how this research could inform mental health interventions tailored to disadvantaged communities would make the abstract more impactful.

Thank you for pointing this out. We have included the statement about how this research could inform mental health interventions tailored to disadvantaged communities

Line 67-75: These findings emphasize the need for targeted mental health interventions and support systems tailored to vulnerable communities. These interventions could inform policies aimed at strengthening mental health services, fostering adaptive coping strategies and promoting help-seeking behaviours among adolescents in socio-economically challenged setting.

2. 2. Introduction

• To strengthen the relevance, consider providing a brief overview of the unique characteristics of the PPR communities (e.g., socio-economic constraints, limited access to mental health resources). This would contextualize the challenges faced by adolescents in these settings and reinforce the importance of the study.

Line 96-103: The dense, crowded and low-resource setting predominantly experienced in PPR communities, often means that children based in these living environments are more susceptible to socio-economic deprivations, and are at higher risk of developing mental health conditions.

• While the authors have referenced international studies on adolescent coping and help-seeking, the literature review could benefit from further discussion of any regional or cultural factors specific to Southeast Asia or Malaysia, which might influence coping strategies and help-seeking behaviors. This would provide additional context for the study’s relevance

Thank you for your valuable feedback.

We have highlighted this suggestion in the revised manuscript.

Kindly please refer to line 148-153

• Consider explicitly stating the research question or hypothesis at the end of the introduction. This would give readers a clear understanding of the study’s focus and objectives.

Kindly please refer to line 157 to 160

3. 3. Methods

• A more detailed explanation of the coding and analysis process would improve transparency. Specifically, it would be beneficial to describe how coding was conducted (e.g., use of software, independent coding by multiple researchers) and how disagreements were resolved to ensure inter-coder reliability.

Kindly please refer to line 245-262

• Additionally, mentioning whether the researchers used any form of reflexivity to

minimize bias would strengthen the methodological robustness.

Thank you for pointing this out. We have included the statement about how used reflexivity to minimize researchers’ biases;

Kindly please refer to line 209-213

In this study, reflexivity was used as a strategy to reduce researcher bias by reflecting on how personal perspectives, assumptions and beliefs could influence data analysis and interpretation. Debriefing and research team member checking were employed as tools to critically examine the researcher's role and potential biases throughout the research process.

4. 4. Results

• Including some quantitative data, such as the proportion of participants endorsing each coping strategy or barrier, would provide additional context and help quantify the prevalence of specific behaviors.

Thank you for the suggestion regarding the inclusion of quantitative data to provide additional context. However, this study is rooted in a qualitative methodology, which is designed to explore in-depth,and rich insights rather than quantify behaviours or experiences. As such, quantifying the proportion of participants endorsing specific coping strategies or barriers would not align with the principles of this qualitative approach.

Instead, we have prioritized thematic analysis to highlight the recurring patterns and variations in coping strategies and barriers among participants. This approach allows us to provide detailed and contextualized understandings of the lived experiences of adolescents in PPR communities. While we do not present numerical prevalence, we ensure transparency and rigor by supporting our findings with representative quotes and robust thematic evidence.

• Furthermore, a summary table outlining each theme with supporting quotes could enhance the visual presentation of the findings, making the results more accessible to readers.

Thank you for your suggestion. Kindly refer to table 2

5 5. Discussion

• The authors could further expand on how their findings might influence specific public health or educational policies, especially in low-resource settings, and suggest avenues for future research Thank you for your valuable feedback.

We have highlighted this suggestion in the revised manuscript;

Kindly refer to line 612 to 619.

As for future research, kindly refer to line 658 to 667.

“Future research on adolescents' mental health during and after the COVID-19 pandemic should consider several directions to overcome limitations identified in current studies. Longitudinal studies are essential to track changes in coping strategies and help-seeking behaviours over time, offering insights into long-term effects. Intervention studies tailored to vulnerable communities, like psycho-education and resilience training, can test effectiveness in promoting adaptive coping strategies. Comparative studies between adolescents in different community types such as urban, rural and affluent settings can highlight unique challenges, aiding targeted intervention design. Community-Based Participatory Research (CBPR) involving adolescents, parents, educators, and local organizations ensures culturally relevant interventions and builds trust.”

6 6. Conclusion

• A brief statement highlighting the potential for community-based mental health programs tailored to the PPR context would provide a forward-looking end to the paper, emphasizing actionable insights that policymakers and practitioners can take from this research. Kindly refer to line 692 to 699

By highlighting the coping strategies and barriers to help-seeking behaviours, the findings provide valuable insights into how adolescents in low-resource settings navigate psychological distress. These findings emphasize the critical need for community-based mental health programs tailored to the PPR context. Such initiatives could focus on fostering adaptive coping strategies, reducing stigma around mental health, and improving access to support services. Policymakers and practitioners can leverage these insights to develop targeted interventions that promote mental health and resilience among vulnerable adolescents, ensuring their well-being in both times of crisis and beyond.

---

## [Editor Report · Decision Letter 2]

3 Jan 2025

PONE-D-23-42611R2Noteworthy trends in maladaptive coping strategies and hindrances to help-seeking behaviour among adolescents living in Malaysia's People’s Housing Project (PPR) during the COVID-19 pandemic: a qualitative studyPLOS ONE

Dear Dr. Subramaniam,

Thank you for submitting your manuscript to PLOS ONE. After careful consideration, we feel that it has merit but does not fully meet PLOS ONE’s publication criteria as it currently stands. Therefore, we invite you to submit a revised version of the manuscript that addresses the points raised during the review process.

These are revisions from me the Academic Editor. I see that you have answered all the queries of the last Reviewer, however, the underlisted remain before the paper can be acepted. They are editing tasks and I encourage you to work on them. Thank you.

REVISIONS AUTHOR NEEDS TO WORK ON—MOSTLY  EDITING

FOR THE REVISED SECTIONS

P. 2, line 68: **This findings** study emphasize the need for

p. 5, Line 144: Additionally, in the context of Asian nations and low and middle income countries, 144 cultural factors such as stigma surrounding mental health **issue (** 16), familial support systems 145 (17) and awareness of mental health (18) may further shape adolescents’ coping strategies 146 and help-seeking behaviours. Addressing these cultural significance is critical to 147 understanding and effectively intervening in mental health challenges faces by this 148 population. 149 150 Hence, th

P. 8, line 245: coding framework. Within these discussions, **the ** research team also identified and incorporated (YOU OMITTED ‘THE’)

FOR THE ORIGINAL SECTIONS THAT WERE NOT REVISED

P. 22, line 612: provide objective outcomes which can further **limits ** the generalizability of current findings (SHOULD BE ‘LIMIT’)

P. 22, line 617: Secondly, the presence of the researcher and counsellor during the **(OMISSION HERE)** may affect

P. 22, line 618: participants’ responses which **may leading** to biased results (SOMETHING IS NOT RIGHT WITH ‘MAY LEADING’….

We look forward to receiving your revised manuscript.

Kind regards,

Adobea Yaa Owusu, MA, PhD, MPH

Academic Editor

PLOS ONE

Journal Requirements:

Additional Editor Comments:

None

Reviewers' comments:

NONE

---

## [Author Response · Author response to Decision Letter 3]

13 Jan 2025

No Comments Feedback

1.

P. 2, line 68: This findings study emphasize the need for Thank you for your valuable feedback.

Line 67: These findings

p. 5, Line 144: Additionally, in the context of Asian nations and low and middle income countries, 144 cultural factors such as stigma surrounding mental health issue (16), familial support systems 145 (17) and awareness of mental health (18) may further shape adolescents’ coping strategies 146 and help-seeking behaviours. Addressing these cultural significance is critical to 147 understanding and effectively intervening in mental health challenges faces by this 148 population. 149 150 Hence, th Line 161: Health issues

P. 8, line 245: coding framework. Within these discussions, the research team also identified and incorporated (YOU OMITTED ‘THE’) Line 267: The research team

P. 22, line 612: provide objective outcomes which can further limits the generalizability of current findings (SHOULD BE ‘LIMIT’) Line 749: limit the generalizability

P. 22, line 617: Secondly, the presence of the researcher and counsellor during the (OMISSION HERE) may affect Line 754: Secondly, the presence of the researcher and counsellor during the IDIs may affect

P. 22, line 618: participants’ responses which may leading to biased results (SOMETHING IS NOT RIGHT WITH ‘MAY LEADING’….

Line 754- 755:

Secondly, the presence of the researcher and counsellor during the IDIs may affect participants’ responses.

which may leading to biased results (OMITTED)

We have addressed all the grammatical issues raised by the reviewer - additionally we detected and corrected several others - as shown by track changes in the revised version of the manuscript.

---

## [Editor Report · Decision Letter 3]

15 Jan 2025

Noteworthy trends in maladaptive coping strategies and hindrances to help-seeking behaviour among adolescents living in Malaysia's People’s Housing Project (PPR) during the COVID-19 pandemic: a qualitative study

PONE-D-23-42611R3

Dear Dr. Subramaniam,

We’re pleased to inform you that your manuscript has been judged scientifically suitable for publication and will be formally accepted for publication once it meets all outstanding technical requirements.

Kind regards,

Adobea Yaa Owusu, MA, PhD, MPH

Academic Editor

PLOS ONE

Additional Editor Comments (optional):

None

Reviewers' comments:

None

---

## [Editor Report · Acceptance letter]

PONE-D-23-42611R3

PLOS ONE

Dear Dr. Subramaniam,

I'm pleased to inform you that your manuscript has been deemed suitable for publication in PLOS ONE. Congratulations! Your manuscript is now being handed over to our production team.

Kind regards,

on behalf of

Professor Adobea Yaa Owusu

Academic Editor

PLOS ONE